# Evaluating the Impacts of Climate Factors and Flavonoids Content on Chinese Prickly Ash Peel Color Based on HPLC-MS and Structural Equation Model

**DOI:** 10.3390/foods11162539

**Published:** 2022-08-22

**Authors:** Tao Zheng, Ding-Ling Zhang, Bing-Yin Sun, Shu-Ming Liu

**Affiliations:** 1College of Science, Northwest Agriculture and Forestry University, Yangling 712100, China; 2Yangling Vocational & Technical College, Yangling 712100, China

**Keywords:** Chinese prickly ash (*Zanthoxylum bungeanum* Maxim.), flavonoids content, peel color, structural equation model, climatic driving

## Abstract

Climate affects Chinese prickly ash peel color directly through temperature and illumination and indirectly influences it through its effect on flavonoid compounds. In this study, a comprehensive evaluation strategy based on high performance liquid chromatography-mass spectrometry (HPLC-MS) technology and a structural equation model was applied to evaluate the effects of climate factors and flavonoids on Chinese prickly ash peel color. There were obvious geographical variations of peel color and flavonoid compounds with an obvious east–west distribution trend which were divided into high-altitude type and low-altitude type. Through path analysis, the wind speed, temperature and annual sunshine duration were found to be the key environmental factors affecting the flavonoids content and peel color, and their direct effects were higher than their indirect effect. Based on HPLC-MS technology and a structural equation model, correlation models of climatic factors and flavonoids with peel color were established, and the factors that had greater weight on pericarp color were obtained. Our results provide experimental evidence that climate factors affect the peel color by affecting flavonoid biosynthesis and accumulation, reveal the geographical variation of peel color and flavonoid component contents in Chinese prickly ash peel, establish a quantization color method for rapid evaluation of peel quality, expand on the influence of climatic factors on flavonoids content and peel coloration and promote agricultural practice in areas with similar climatic conditions.

## 1. Introduction

*Zanthoxylum bungeanum* Maxim., known as Chinese prickly ash, is widely distributed in China and was the most important, traditional condiment and medicinal plant in China for hundreds of years and has strong ecological adaptation [1,2,3]. Due to the influence of other species, geographical conditions, climate factors and chemical composition, the color of Chinese prickly ash peel produced in different regions is different [4]. The main chemical components involved in Chinese prickly ash peel color are flavonoids, volatile oil, alkaloids, amides, and so on, of which flavonoids are natural pigments with good stability. Through research into the application of Chinese prickly ash, it was found that it is used as a traditional Chinese medicine to treat toothache, eliminate colds, stop pain and increase appetite; it is also used for antibacterial and insecticidal purposes [5,6]. It is also used as a good condiment with a unique tingling taste for cooking food [7].

Flavonoids are widely distributed in plants and are the secondary metabolites produced during long-term natural selection which have some therapeutic effects such as antioxidant, antiischemic, anticancer, antiinflammatory and antibacterial effects [8,9]. More and more flavonoid and anthocyanidin compounds have been identified from the Chinese prickly ash, such as rutin, hyperoside, quercetin, quercitrin and anthocyanidins [10,11]. Quercetin has the strongest anticancer, antiinflammatory and anti-heart-disease effects [12]. Hyperoside possesses various functions, such as antiinflammatory, antibacterial, antiviral and antitumor functions [13]. Anthocyanin is a flavonoid compound that produces purple and red color in plants. Anthocyanins, with strong antioxidant capacity, have been widely recognized by consumers as health food raw materials. Natural flavonoids extracted from plants with high purity and strong activity can be used for processing special functional health food and medicines [14]. Recently, foreign countries have paid great attention to the development of flavonoids as food additives, coke drinks, chewing gum, bread and beer, and other foods with flavonoids have been developed. These foods, without preservatives, taste good and have certain antibacterial and bactericidal effects. Color is the main index for quality evaluation of Chinese medicinal materials, and it is also related to the content of internal components. Plants are rich in flavonoids, and flavonoids provide color pigments for some parts of plants, such as petals and peel. The color of mature peel is mainly controlled by flavonoids. A difference in color often reflects a difference in the quality of medicinal materials, but there is great subjectivity in color evaluation. In recent years, the number of investigations into the correlation between peel color and material components has gradually increased, and the colorimeter method can be used to quantitatively and objectively reflect the internal quality, which was confirmed in *Carthamus tinctorius* [15], *Lonicera japonica* [16], *Rubus chingii* [17], *Phellodendron chinense* [18], *Morus alba* [19] and other medicinal materials.

Fruit color is divided into ground color and cover color, of which the ground color is mainly regulated by flavonoids and other pigments, and the cover color is mainly controlled by anthocyanins. The ground color and cover color determine the diversity of fruit color. The development of fruit color is regulated by flavonoids, anthocyanin and various climate factors [20]. Moreover, there are obvious geographic differences in the types and contents of chemical constituents in medicinal resources. Recently, climate factors’ effects on bioactive metabolite content in food and medical herbs (including asafoetida (*Ferula assa-foetida*), kale (*Brassica oleracea* var. *sabellica*), bilberry (*Vaccinium myrtillus*), turmeric (*Curcuma longa*) and *Sinopodophyllum hexandrum*) in different geographical locations have been reported [21,22,23,24,25]. Flavonoids and anthocyanins are pigment compounds that are sensitive to environmental conditions, and their synthesis is regulated by temperature, oxygen and light. Åkerström found that anthocyanins content in bilberries was higher in berries from the northern regions than in those from the southern regions of Sweden [26]. Studies have also reported that altitude seems to influence bilberry anthocyanin content [27]. Katerina Biniari found that the accumulation of flavonoids in grape skins and seeds is controlled and regulated by air temperature and wind speed [28]. The characteristics of flavonoids and phenolic compounds in Chinese prickly ash pericarps are widely variable due to irregular adaptation to environmental conditions [29]. Granato’s studies revealed that the component, in turn, reacts strongly to the environment in which the trees are grown [30]. However, studies on Chinese prickly ash have mainly focused on the genetic diversity, the contents of volatile oil numb-taste components and their antioxidants [31,32]. There is a lack of comprehensive and systematic investigation into geographical variations of peel color and their correlations with climatic factors and flavonoids. Therefore, a more comprehensive and effective strategy should be applied to evaluate the impacts of climate factors and flavonoid compounds on Chinese prickly ash peel color from different habitats.

Here, 26 Chinese prickly ash peels samples collected from their natural distribution areas in China were selected as experimental materials, and the effects of flavonoids content and climatic factors on peel color were explored by using statistical methods such as a structural equation model. The chemotypes of Chinese prickly ash from different producing areas were classified based on the flavonoids content in the peels. The response characteristics of climatic factors and peel color were systematically studied based on average annual climatic factors and multivariate statistical methods used to explore the key climate factors causing peel color geographical variation. The correlation between peel color and main flavonoids content was investigated to provide a new method for quality evaluation. The results may contribute to a better understanding of the response characteristics between climate factors, flavonoid compounds and peel color, find out the climatic causes of the geographical variation of peel color, reveal the ecological mechanism of the formation of different chemical ecotypes and provide guidance for agricultural climate zones with high quality suitable for large-scale cultivation of Chinese prickly ash.

## 2. Materials and Methods

### 2.1. Plant Materials and Chemicals

The 26 Chinese prickly ash peels materials were collected from eight provinces of China (the main production areas: Guide, Xunhua, Hanyuan, Jiuzhaigou, Wenxian, Wudu, Qin’an, Fengxian, Fuping, Hengshan, Hancheng, Yongji, Lingbao, Jiaocheng, Shexian, Zaozhaung and Laiwu) at different altitudes (201–2188 m) from July to August 2020 (Figure 1). The 26 samples were divided into 6 groups based on the geographical location and altitude of the sample collection sites, as shown in Appendix A. Under the premise of protecting local germplasm resources, Chinese prickly ash mature fruits from each location were collected in three replicates, and the distance between each plant was more than 50 m. Fruits without pests and mechanical damage were dried in the laboratory at room temperature until they reached a constant weight.

A total of 15 flavonoid compound standards (hyperoside, luteolin, kaempferol, quercitrin, catechin, rutin, chlorogenic acid, quercetin, hesperetin, apigenin, peonidin O-hexoside, peonidin 3-O-glucoside, cyanidin 3-O-glucoside, cyanidin O-syringic acid, cyanidin 3-O-galactoside) were bought (Beijing Solarbio Science & Technology, Beijing, China). HPLC-grade methanol, acetic acid and acetonitrile were bought (TEDIA Chemical Co., Ltd., Fairfield, OH, USA). Deionized water (18 MΩ cm) was used to prepare aqueous solutions (MilH-Q Advantage A1, Millipore, Billerica, MA, USA).

### 2.2. Determination of Color Quality

The fresh fruits were placed and filled in a glass-surface vessel, and the peel color was measured using a NH310 computer colorimeter (Shenzhen ThreeNH Technology Co., Ltd., Shenzhen, China). Each sample was photographed five consecutive times, the chromaticity values, such as “L*” (lightness), “a*” (greenness to redness) and “b*” (blueness to yellowness), were recorded and the average values were obtained after determination.

### 2.3. Sample Preparations

The dried peels were ground to a powder and sieved through a no. 60 mesh (<0.250 mm). Each sample was accurately weighed at 1.0 g and extracted with 30 mL of 70% methanol at 50 °C for 40 min by ultrasonication and then centrifuged at 15,000 rpm for 10 min. The supernatant was filtered by microporous membrane (SCAA-104, 0.22 µm pore size, ANPEL, Shanghai, China), and the filtrates were stored in the injection bottle.

### 2.4. HPLC-MS Analysis of Fifteen Flavonoids Compounds

The quantitative analysis of the flavonoid compounds in the 15 Chinese prickly ash peels was carried out based on HPLC-MS (HPLC, Shim-pack UFLC SHIMADZU CBM30A system, Shimadzu, Kyoto, Japan; MS, Applied Biosystems 4500 Q TRAP, Thermo Scientific)) technology. The HPLC conditions were as follows: column, Waters ACQUITY UPLC HSS T3 C18 (1.8 µm, 2.1 mm × 100 mm); solvent system: solvent A, water (0.04% acetic acid), solvent B, acetonitrile (0.04% acetic acid); gradient program, 95:5 *v*/*v* (A/B) at 0 min, 5:95 *v*/*v* at 11.0 min, A:95 *v*/*v* at 12.0 min, 95:5 *v*/*v* at 12.1 min, 95:5 *v*/*v* at 15.0 min; flow rate, 0.40 mL/min; column temperature, 40 °C; injection volume, 5 μL. The effluent was alternatively connected to an ESI-triple quadrupole-linear ion trap (QTRAP)-MS. The ESI source operation parameters were as follows: ion source, electrospray ion source; scan range, 100–600 *m/z*; source temperature, 550 °C; ion spray voltage (IS), 5500 V; curtain gas, 25 psi.

Flavonoid data analysis was performed by Analyst 1.6.3 software (AB SCIEX, Concord, Concord, ON, Canada). Integration and correction of chromatographic peaks were performed using MultiQuant 3.02 (AB SCIEX, Concord, ON, Canada). The corresponding relative flavonoids content was expressed as the chromatographic peak area integral.

### 2.5. Data on Climate Factors

A total of 10 climate factors (temperature, relative humidity, wind speed, sunshine duration, precipitation, sunshine percentage) of each sampling site were provided by Yangling Meteorological Administration. The detailed information of the 10 climate factors is listed in Appendix A.

### 2.6. Data Analysis

Chemometric analyses, such as heatmap analysis, principal component analysis (PCA), correlation analysis, orthogonal partial least squares-discriminant analysis (OPLS-DA), regression analysis and path analysis, were used to reveal the flavonoids and climate factors’ effects on the Chinese prickly ash peel color. Cluster analysis, PCA and OPLS-DA were carried out using R (http://www.r-project.org/ (accessed on 20 May 2022)). Path analysis and regression analysis were performed using SPSS 24.0 software (SPSS Inc., Chicago, IL, USA). A structural equation model was established to analyze the relationship between peel color and climatic factors and flavonoids by the lavaan package in R (http://www.r-project.org/ (accessed on 25 May 2022)).

## 3. Results

### 3.1. Color Determination of Chinese Prickly Ash Peels and Their Geographical Variation

CIE (International Lighting Commission) results revealed that Chinese prickly ash peel color varies greatly in different habitats. The L*, a* and b* value parameters of Chinese prickly ash peel color from different habitats are presented in Figure 2. The L* values representing the lightness of all fruit samples from different habitats were determined; the L* values were between 73.398 and 92.052, and the fruit color lightness was higher in the high-altitude area. The color index a* values were higher than the b* values in the high-altitude area, indicating that the dominant peel color was red, and, in the low-altitude area, the opposite. Among them, the high a* value in the peels from the high-altitude area could be seen, with values greater than 133.46, whereas low a* values were recorded in the peel from the low-altitude area. The mean a* value gradually increased from the eastern region to the northwestern region, displaying a geographic group variation trend. However, this geographical group variation was not a simple, continuous variation, and Chinese prickly ash peel samples were obviously divided into two ecological-geographical groups. From the east to the west, the ratio of yellow fruit gradually decreased, and the ratio of red fruit gradually increased. In the western high-altitude region, the color of the fruit was redder, and the peel color was slightly yellow in the mid-eastern, low-altitude area. Therefore, the Chinese prickly ash peel color from different habitats has regional differences.

### 3.2. Quantification of the Fifteen Flavonoid Compounds of 26 Chinese Prickly Ash Peels

The reliable and replicable HPLC-MS method was applied to the simultaneous determination of the fifteen flavonoid components in 26 populations of Chinese prickly ash peels (Appendix A). Flavonoid compounds were extracted and analyzed in triplicate (summarized in Table 1). Wide variation and significant differences (*p* < 0.05) were observed in the fifteen flavonoid components in the Chinese prickly ash peels. The hyperoside and quercitrin contents, which are the main flavonoids compounds in Chinese prickly ash peels, were significantly higher than that of other compounds, and Y_CGC_ (cyanidin 3-O-glucoside) and Y_CGT_ (cyanidin 3-O-galactoside) were the predominant anthocyanidin. Y_QI_ was the most abundant component, and the content varied from 7.55 ± 0.42 to 45.13 ± 0.06 mg/g. The next most plentiful compounds were Y_HY_ (6.03 ± 0.08 to 27.84 ± 0.38 mg/g), Y_CA_ (2.64 ± 0.28 to 21.06 ± 0.21 mg/g), Y_C_ (4.65 ± 0.16 to 13.55 ± 0.62 mg/g), Y_RU_ (3.28 ± 0.19 to 13.07 ± 0.11 mg/g), Y_PH_ (5.34 ± 0.15 to 12.13 ± 0.22 mg/g) and Y_PG_ (4.48 ± 0.17 to 11.58 ± 0.46 mg/g). The content of Y_LU_ (0.29 ± 0.02 to 3.80 ± 0.28 mg/g), Y_KP_ (0.66 ± 0.04 to 2.99 ± 0.04 mg/g) and Y_QU_ (0.26 ± 0.03 to 5.16 ± 0.38 mg/g) was lower. The Y_QI_, Y_QU_, Y_LU_, Y_CA_ Y_CGC_ and Y_CGT_ in different regions also had obvious geographical variations, among which the contents of the four above components in peels from a high altitude were higher than in those from a low altitude. However, Y_KP_ and Y_C_ displayed no obvious regional differences (Table 1). The results of this study exhibited that the content of fifteen flavonoids in the Chinese prickly ash peels under different climatic conditions was uneven, indicating that differences in flavonoids content in Chinese prickly ash peel might be attributed to its geographic locations and chemotype.

### 3.3. PCA and HCA Analysis

To further investigate the geographical variation of flavonoid components in Chinese prickly ash peel from different provenances, we explored the correlations between flavonoid compounds and the geographic factors of the sampling sites using Pearson’s correlation coefficient (Figure 3A). Except for Y_HE_, Y_C_ and Y_RU_, the remaining flavonoids were positively correlated with altitude and wind speed and strongly positively correlated with altitude (*p* < 0.01). They were negatively correlated with X_AAP_, X_ASD_ and X_AMT_ and were significantly negatively correlated with longitude (*p* < 0.05). The flavonoids content increased against altitudinal gradients and decreased with the increasing longitude (Figure 3A). There was obvious regional differentiation between the east and the west, and the flavonoids contents in the west was higher than in the east.

Principle component analysis (PCA) was used to reveal the accumulation differences and variability of flavonoids compounds among peels from different habitats. In our research, two principal components (PC1 and PC2) were extracted, which accounted for 48.36% and 14.59%, respectively (Figure 3B). Moreover, the cumulative contribution rate reached 62.95%. There was a trend that the twenty-six peel samples were separated as relatively independent, and the peel samples were divided into four groups. In the PCA 2D map, the sample clustering could be seen more intuitively. Through principal component analysis, it was found that the difference in flavonoid compounds among samples might be the difference among varieties from different habitats. Orthogonal signal correction and partial least squares-discriminant analysis (OPLS-DA) was an effective method for maximizing the difference between groups (Figure 3C). The results of OPLS-DA demonstrated that those samples were obviously assigned to two categories and four groups, consistent with the PCA results.

Based on the unit variance scaling of flavonoid component contents, the heat map was drawn by the ComplexHeatmap package of R software, and hierarchical cluster analysis (HCA) was carried out to analyze the correlation between Chinese prickly ash germplasms from different habitats. The hierarchical cluster analysis of the flavonoids content divided the Chinese prickly ash populations into two categories (Figure 3D). The first cluster was composed of group A and group B, which had higher quercitrin, luteolin and quercetin. The second cluster was generated by germplasm resources in low-altitude areas. The reason for the separation of these two clusters was the different flavonoid component content in the peels. A1 (Guide, China), A2, A3 (Xunhua, China), A4, A5 (Hanyuan, China) and A6 (Jiuzhaigou, China) were grouped together. All of them came from high-altitude habitats. Samples B1, B2, B3, B4 and B5 were divided into a group. These samples mostly came from Gansu and Shanxi provinces. Samples E1, E3, E4 and E5 were clustered into a larger group; all of them came from Shandong province. The last group contained the remaining samples, which largely came from Shanxi, Shaanxi and Henan provinces and were located in the intersection of Henan, Shaanxi and Shanxi provinces. Based on the results of cluster analysis and flavonoid compound content, some geographical provenances in group A and group B could be used as cultivation bases for flavonoids-rich Chinese prickly ash. Geographically, the variation of flavonoids in Chinese prickly ash populations presented an obvious east–west trend, and the groups with similar geographical distance could be clustered into one group at the altitude gradient, indicating that the variation of chemical composition content in fruit at altitude was continuous. Moreover, the flavonoids contents in peels from Fengxian of Shaanxi, Wudu and Qin’an of Gansu were higher, which could be used as the origins of the excellent Chinese prickly ash germplasm resources.

### 3.4. Direct and Indirect Effects of Climate Factors on Flavonoid Compounds and Peel Color

Based on the climatic factors of different regions, the response relationship between flavonoid compounds and climatic factors was analyzed successively. The results of the correlation analysis (Figure 3A and Table 2) demonstrated that the flavonoid compounds were correlated with all the climate factors but to different degrees. The correlation analysis between climatic factors is summarized in Appendix A. Most compounds were negatively correlated with X_AAP_, X_AMT_, X_AMAT_, X_AMIT_, X_ASD_ and X_RH_. Y_HY_, Y_PG_, Y_CGC_ and Y_CSA_ were significantly negatively correlated with X_AMT_ and X_AMAT_, and Y_HY_ was also significantly negatively correlated with X_AMIT_ and X_RH_ (*p* < 0.01). Y_HY_ was significantly positively correlated with X_MW_, X_EW_ and X_MAW_ (*p* < 0.05), and Y_QI_, Y_CA_, Y_PG_ and Y_CGC_ were significantly positively correlated with X_MAW_ (*p* < 0.05). X_ASD_ was positively correlated with Y_HY_ (*p* < 0.05), and X_ASP_ was significantly positively correlated with Y_HY_ (*p* < 0.01). X_AAP_ was significantly negatively correlated with Y_HY_ and Y_CSA_ (*p* < 0.01) and negatively correlated with Y_CGC_ (*p* < 0.05). The results of the correlation analysis indicate that regions with low temperature, low precipitation and high wind speed are suitable for the production and accumulation of flavonoid compounds in Chinese prickly ash peels.

Path analysis (PA) was performed to gain insight into the direct and indirect effects of climate factors on flavonoid compounds. Ten climate factors were used as dependent variables, and 15 flavonoids and peel color values were selected as independent variables to carry out the analysis. Stepwise regression analysis was performed on the climate factors, flavonoids and peel color by SPSS 24.0 software. Then, based on the regression analysis (Appendix A), the dominant climate factors of each compound were screened out, and, finally, the direct path coefficients and indirect path coefficients were calculated. The results of the PA (Table 3) in this study demonstrated that the effects of climate factors on flavonoid compound contents are significant. For hyperoside, the negative direct effect on the climate factors was X_RH_, with a coefficient of −0.720. X_ASD_ showed significantly negative direct effects and positive indirect effects on Y_LU_, Y_QI_, Y_CA_, Y_AP_, Y_PH_ and Y_PG_ (*p* < 0.05). X_AMAT_ played a negative direct and positive indirect role in the accumulation of Y_QI_, Y_AP_, Y_PH_ and Y_PG_ and exhibited a negative direct role in the accumulation of Y_CGC_ and Y_CSA_. X_MAW_ was the key environmental factor for Y_QI_, Y_PH_ and Y_CA_ and was present with positive direct effects. X_MW_ exhibited a negative direct effect and a positive indirect effect on Y_PH_. X_ASP_ displayed a positive direct effect and negative indirect effect on Y_LU_, Y_QI_ and Y_CA_. X_ASD_ and X_MW_ had the most positive indirect effect (0.384, 0.615) on Y_PH_, but with a negative direct effect (−0.700, −0.470) and the lowest correlation coefficient (−0.316, 0.145), which suggests that the indirect effect of X_ASD_ and X_MW_ on Y_PH_ was the contributory cause of relevance. For a*, X_AMAT_ and X_ASP_ displayed a negative direct and indirect effect on a*, but X_MAW_ had a positive direct and indirect effect. X_AMAT_ showed significantly negative direct effects on L*(*p* < 0.01). X_AMAT_, X_AMT_ and X_AAP_ were the key climate factors for b*.

The decision coefficient is a decision index in path analysis. It was used to rank the comprehensive effect of climatic factors on the accumulation of flavonoid compounds and determine the main climatic factors that mainly affect their synthesis. The positive value of the decision coefficient indicated that climate factors can promote the accumulation of flavonoids, and the negative value of the decision coefficient indicated that climate factors can inhibit the accumulation. Combined with the results of correlation analysis and path analysis, temperature, sunshine duration, wind speed and precipitation were strongly correlated with the flavonoid compound contents. By analyzing the climate differences of climate factors in areas of Chinese prickly ash with different quality, the climate characteristics were obtained. Combined with the correlation between the 15 flavonoids components and the 10 climate annual factors, the flavonoids accumulation was suitable in areas with low temperature, less precipitation and strong wind speed. Based on the differences in temperature, sunshine and precipitation between high altitudes (Qinghai, Gansu and western Shaanxi province) and low altitudes, there were significant differences in flavonoids contents in Chinese prickly ash peels from different regions.

### 3.5. Correlation between Peel Color and Flavonoid Compounds

The correlation between flavonoids contents and color values was analyzed (Table 2). The results showed that the color values L* and a* were significantly positively correlated with flavonoids content (*p* < 0.05), and b* was negatively correlated with flavonoids content (*p* < 0.05). Y_C_ presented a significantly positive correlation with L* and a* (*p* < 0.05). Y_CA_ exhibited a significantly positive correlation with a* and a significantly negative correlation with b* (*p* < 0.05). Y_LU_, Y_QI_, Y_AP_, Y_PH_, Y_PG_, Y_CGC_, Y_CSA_ and Y_CGT_ were significantly positively correlated with L* and a* values and were significantly negatively correlated with b* (*p* < 0.01). The results indicated that the higher a* and lower b*, the redder the peel color, and the higher the contents of anthocyanins, apigenin and quercitrin.

The flavonoids content was used as the dependent variable, and L*, a* and b* were used as independent variables for regression analysis to explore the quantitative relationship between the content and color value. The R^2^ values of flavonoids content and color indexes (a* and b*) were greater than 0.653, indicating that contents of these substances above 65.3% are reflected by color values (Table 4). The above compounds content had significant regression with the color values a* and b*, which indicated that a method for predicting flavonoids content in Chinese prickly ash peels was found by quantifying the peel color value combined with the regression equation.

### 3.6. Structural Equation Models of the Effects of Climate Factors and Flavonoids on Peel Color

The structural equation model (SEM) offers a means to evaluate hypothesized causal relationships amongst multiple variables. Based on this existing understanding, the conceptual model, as shown in Figure 4A, was first established in the study area using the peel color, flavonoids and climatic factors as driving data, in which the latent variables included climatic factors and flavonoids. The lavaan package in R was used to establish the SEM to analyze the direct and indirect effects of climatic factors and flavonoids on peel color. Using sample data for model fitting analysis, CMIN/DF < 1, RMSEA < 0.08 and AGFI > 0.9 indicated that the model was excellent with reasonable adaptation. Therefore, the models with CMIN/DF < 1, RMSEA < 0.08 and AGFI > 0.9 were selected. For peel lightness, the selected SEM explained 68% of the L* variation in Chinese prickly ash peels (Figure 4B). Y_PG_ had a significant direct effect on peel color brightness, and the standardized path coefficient was 0.75 (*p* < 0.01). The X_MAW_ had a direct impact on L* with a standardized path coefficient of 0.21, which also indirectly affected L* through Y_PG_, and the standardized indirect path coefficient was 0.34 (0.45 × 0.75). X_AMT_ and X_AMAT_ influenced L* indirectly via Y_PG_ with indirect path coefficients of −0.49 (0.66 × 0.75) and −0.34 (−0.45 × 0.75). The SEM explained 66% of the total variation in a* (Figure 4C). X_AMAT_, X_MAW_ and Y_CGC_ had a direct impact on a*, and the standardized path coefficients were −0.46 (*p* < 0.05), 0.36 (*p* < 0.05) and 0.88 (*p* < 0.01), respectively. The standardized path coefficients of X_AMAT_, X_MAW_ and Y_CGC_ were −0.50 and 0.41 (*p* < 0.05), respectively. It could be seen that X_AMAT_ and X_MAW_ not only directly affected peel color, but also exerted indirect influence via Y_CGC_. The indirect path coefficients of X_AMAT_ and X_MAW_ were −0.44 (−0.50 × 0.88) and 0.36 (0.41 × 0.88), respectively. The SEM explained 69% of the total variation in b* (Figure 4D). X_MW_ had a negative effect on b*, but the contribution was low. The standardized path coefficient of X_AAP_ was 0.16, which was higher than X_AMT_ (0.10) and X_AMAT_ (0.06). The standardized path coefficient of Y_QI_ was −0.77 (*p* < 0.01), which was greater than that of the climate factors. X_AMT_, X_AMAT_ and X_MW_ indirectly affected b* through Y_QI_, with path coefficients of 0.34 (−0.44 × −0.77), −0.41 (−0.53 × −0.77) and −0.23 (−0.30 × −0.77), respectively.

The X_MAW_ and X_MW_ were positively correlated with flavonoids; the anthocyanins especially showed a significant negative correlation with X_AMT_ and X_AMAT_ in the growing season, indicating that regions with low temperature and high wind speed are conducive to flavonoid formation and accumulation, which indirectly affect the peel color. We found clear evidence that climate factors influence Chinese prickly ash peel color both directly and indirectly (through flavonoids), and the SEM results revealed that the direct effect of climatic factors on peel color is less than the indirect effect, and the indirect effects on peel color are mainly through the flavonoids.

## 4. Discussion

With the increasing planting area and yield of Chinese prickly ash, it is particularly important to monitor and control its quality. As one of the most intuitive external characteristics, peel color is closely linked to its quality. However, the traditional color discrimination method is subjective and susceptible to external factors. Therefore, in this study, the peel color difference was used to convert the subjective index color into objective indexes (L*, a*, b*) using colorimeter method in order to quickly evaluate its quality. Flavonoids are natural pigments, and Chinese prickly ash peel color is related to flavonoids. The anthocyanins contents (peonidin O-hexoside, peonidin 3-O-glucoside, cyanidin O-syringic acid and cyanidin 3-O-galactoside) were significantly positively correlated with L* and a*, indicating that the higher the L* and a* values, the higher the anthocyanins contents.

The geographic difference in chemical compounds content in medicinal plants is a concrete manifestation of the genuineness of Chinese herbal medicines. Climate factors such as precipitation, temperature, wind speed and annual sunshine duration have great influence on the growth and development of medicinal plants and the biosynthesis and accumulation of active ingredients [33,34]. The genetic variation of plant traits has a certain correlation with their geographical distribution [21,35,36]. Zhang et al. revealed that the geographical variation of plants is mainly caused by environmental factors [37]. In the current study, there was an obvious geographical variation in the peel color and flavonoid content of Chinese prickly ash peels from different habitats, and flavonoids contents and peel color were closely related to the altitude with an obvious east–west distribution trend, divided into high-altitude type and low-altitude type. In high-altitude producing areas, the flavonoids contents were relatively high, and the peel color was redder. From the correlation between flavonoids contents, peel color and geographical factors, it can be seen that the best, ecologically suitable area and the best quality formation area are mainly concentrated in high-altitude areas, such as Qinghai, Sichuan, Gansu and western Shaanxi.

Climate factors affect the biosynthesis and accumulation of bioactive metabolites in plants [38]. Olha Mykhailenko found that flavonoid compounds accumulation in Iris species is positively regulated by sunshine duration [39]. Zhang et al. reported that tanshinones and biomass accumulation is affected by average relative humidity and annual mean temperature [35]. We found clear evidence that climate directly and indirectly affects the flavonoid compounds accumulation in Chinese prickly ash peels. For example, the flavonoids contents in three samples (A3, C4, E2), which belonged to the same variety *Zanthoxylum bungeanum* cv. Xiaohongguan but from different regions, varied. It demonstrated that differences in flavonoids contents are probably caused by climate factors. In our study, the relationships between the fifteen flavonoids compounds and climate factors were different. It was suggested that the annual mean temperature, the temperature, wind speed and sunshine duration are the key climate factors for flavonoids compounds contents in Chinese prickly ash peel.

Through the structural equation model, we found clear evidence that climatic factors directly and indirectly (through flavonoids) affect Chinese prickly ash peel color. This supported all the main mechanisms considered here and provided in-depth understanding of their relative importance. In relation to the direct effect of climate on peel color, we found that peel lightness and redness decreased with temperature, suggesting that the variation of peel color is limited by temperature [40]. In relation to the indirect control of climate, we found that the main mechanism of climate factors’ effects on peel color is the indirect pathway mediated by flavonoids [41]. Our results revealed that temperature has a direct effect on peel color, which is consistent with existing evidence. Temperature was described as an important climatic factor in the process of peel coloration, and its effect on anthocyanin synthesis is complex [42]. The large diurnal temperature range in the mature stage is beneficial to flavonoid and anthocyanin accumulation and promotion of fruit coloration. The large diurnal temperature difference in the western region was conducive to the accumulation of carbohydrates and provided the necessary synthetic precursor for the synthesis of anthocyanin. Therefore, the peel color was redder. Light not only affects the synthesis of organic compounds, such as sugar and phenylalanine, but also regulates the activities of enzymes related to anthocyanin synthesis [43]. When the light intensity was more than 70% of the natural light intensity, the peel color was excellent. In the western high-altitude area, sufficient light was conducive to the synthesis of flavonoids, anthocyanins and other substances, so the peel color was brighter and redder. Wind improves aeration and sunlight penetration and accelerates the flow of CO_2_, which is conducive to improving photosynthetic efficiency, promoting sugar accumulation, enhancing the transformation of pigment body to flavonoids and anthocyanins and promoting fruit coloration. The direct and indirect effects of wind speed on peel color are positive, which increase the brightness and red color of fruit. The synthesis and accumulation of flavonoids and anthocyanin are regulated by climatic factors. Low temperature, sufficient light and suitable wind speed are conducive to the peel coloration. In general, the direct effect of climate on peel color is less than the indirect effect affecting flavonoid accumulation. In addition, quercetin might be related to yellow coloration of peel, and two anthocyanins (peonidin 3-O-glucoside, cyanidin 3-O-glucoside) are the key red pigments related to red-colored peel formation.

Given the close relationship between fruit quality and climatic factors in different producing areas, our results demonstrated that climatic factors have indirect effects on peel color from different producing areas through flavonoids, and a method for rapid quality evaluation of peel by quantitative color was established. Our study reflected the climate change on the altitude gradient, and peel color and flavonoids content were divided into high-altitude type and low-altitude type. The high-altitude area had a large diurnal temperature difference and sufficient light, which was conducive to regulating the activity of enzymes linked to anthocyanin biosynthesis, enhancing anthocyanin accumulation and promoting peel coloration. Our study focused on the annual average level of climatic factors from the direct and indirect perspective of causes of Chinese prickly ash peel quality differences to explore the key climatic factors affecting the formation of Chinese prickly ash peels quality. The temperature, sunshine duration, wind speed and precipitation were the main climatic reasons for the quality difference. The effects of soil conditions on peel quality will continue to be considered in subsequent studies. In addition, further studies are needed to assess the associations between climate factors and reproductive and nutritional factors using molecular biology methods and find the causes of the correlation between the active ingredients content and climatic factors in Chinese prickly ash growing areas and comprehensively reveal the ecological mechanism of the genuineness of Chinese prickly ash.

## 5. Conclusions

Climate has an important impact on the Chinese prickly ash peel color, which determines its economic value. The mature fruits’ color is mainly controlled by the flavonoid and anthocyanin type and content, which fluctuates with geographical origin and growth environment. Based on HPLC-MS fingerprint technology and structural equation model, a new strategy was developed to investigate the climate factors’ effects on the Chinese prickly ash peel color. With the established structural equation model, this study successfully established the correlation between climatic factors, flavonoids and peel color, and the main climatic factors affecting the color difference were screened out. The correlation model between ecological factors and flavonoids was established by binary correlation analysis. The results can provide reference for the selection of Chinese prickly ash planting areas and production zoning. Such a method combining HPLC-MS technology with a structural equation model might open up a new strategy for evaluating the effect of climate factors on Chinese prickly peel color and the ecological mechanism of geographical variation in peel color.

## Figures and Tables

**Figure 1 foods-11-02539-f001:**
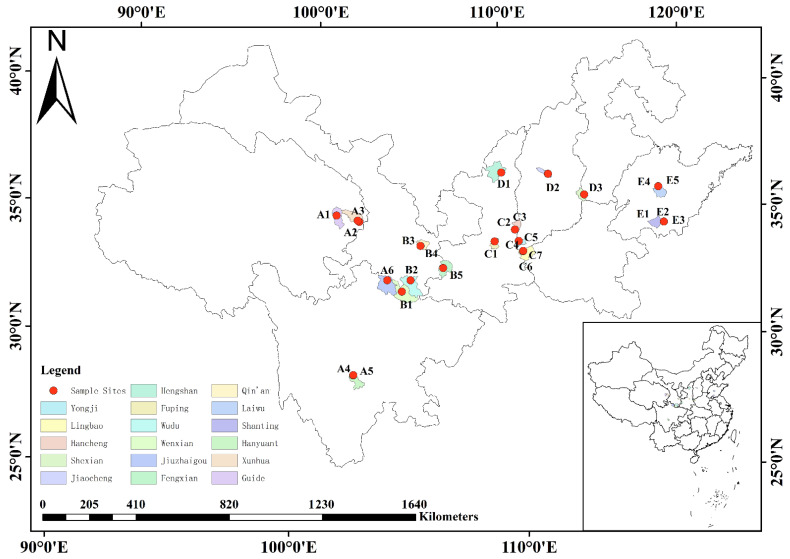
Map of collection sample sites of 26 Chinese prickly ash peels.

**Figure 2 foods-11-02539-f002:**
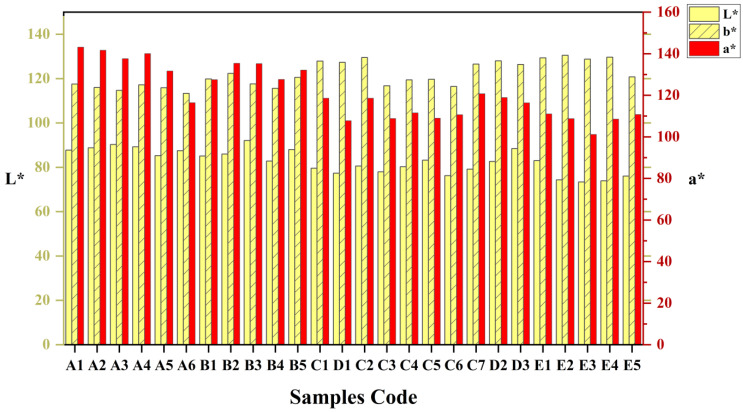
Determination results of color and chromaticity values of Chinese prickly ash peels. L* values represent the brightness of the pixel. The chromaticity coordinates a* and b* denote the position of color in color space. A positive a* value denotes red, and a positive b* value indicates yellow.

**Figure 3 foods-11-02539-f003:**
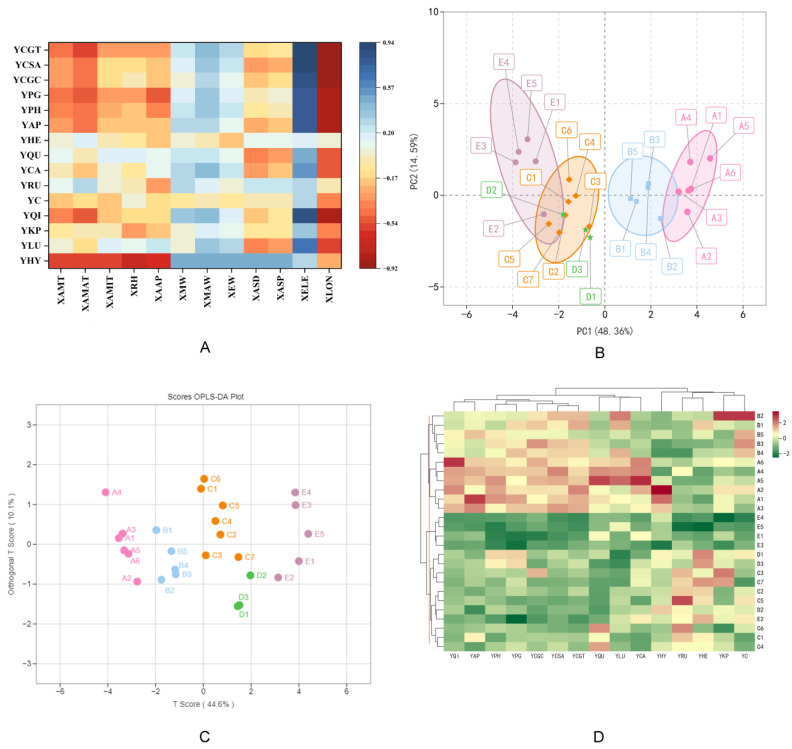
Geographical variation in flavonoid compounds. (**A**): Correlation coefficient analysis between flavonoids content and geographical factors. (**B**): PCA on 2D plot of Chinese prickly ash based on flavonoid compound contents. Four different color ellipses represent PCA results. (**C**): Different flavonoid compounds analysis on the basis of orthogonal signal correction and partial least squares-discriminant analysis (OPLS-DA). (**D**): Cluster heatmap plot of Chinese prickly ash from different origins. Y_HY_—hyperoside, Y_LU_—luteolin, Y_KP_—kaempferol, Y_QI_—quercitrin, Y_C_—catechin, Y_RU_—rutin, Y_CA_—chlorogenic acid, Y_QU_—quercetin, Y_HE_—hesperetin, Y_AP_—apigenin, Y_PH_—peonidin O-hexoside, Y_PG_—peonidin 3-O-glucoside, Y_CGC_—cyanidin 3-O-glucoside, Y_CSA_—cyanidin O-syringic acid, Y_CGT_—cyanidin 3-O-galactoside. X_AMT_—annual mean temperature, X_AMAT_—annual mean maximum temperature, X_AMIT_—annual mean minimum temperature, X_RH_—annual relative humidity, X_MW_—mean wind speed, X_MAW_—maximum wind speed, X_EW_—extreme wind speed, X_ASD_—annual sunshine duration and X_AAP_—annual average precipitation.

**Figure 4 foods-11-02539-f004:**
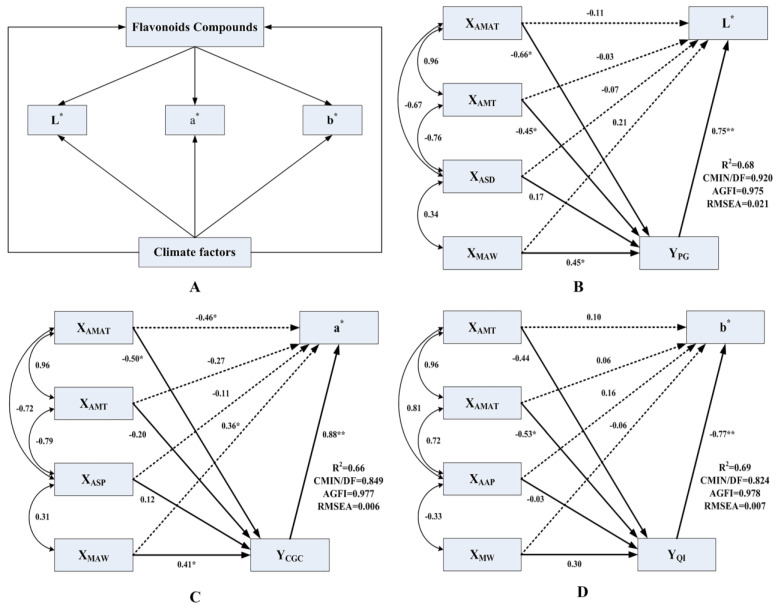
(**A**): The conceptual structural equation modelling was used to examine the linkages amongst climate factors, flavonoids and peel color values. (**B**–**D**): SEM of the effects of climate factors and flavonoids on peel color in L* (**B**), a* (**C**) and b* (**D**). ** represents significant correlation at *p* < 0.01 level, * represents significant correlation at *p* < 0.05 level.

**Table 1 foods-11-02539-t001:** Fifteen flavonoids content (mg/g) in different varietal types of Chinese prickly ash peels samples.

Provenance	Y_HY_	Y_LU_	Y_KP_	Y_QI_	Y_C_	Y_RU_	Y_CA_	Y_QU_	Y_HE_	Y_AP_	Y_PH_	Y_PG_	Y_CGT_	Y_CGC_	Y_CSA_
A1	22.88 ± 0.06 ^b^	2.34 ± 0.06 ^e^	1.89 ± 0.21 ^d^	28.66 ± 0.36 ^e^	7.85 ± 0.02 ^f^	6.42 ± 0.38 ^j^	9.80 ± 0.16 ^f^	3.02 ± 0.28 ^c^	3.46 ± 0.12 ^i^	5.04 ± 0.20 ^a^	6.77 ± 0.13 ^c d^	7.52 ± 0.09 ^b^	12.13 ± 0.22 ^a^	11.58 ± 0.46 ^a^	7.54 ± 0.14 ^e^
A2	27.84 ± 0.38 ^a^	1.92 ± 0.08 ^f^	2.10 ± 0.02 ^c^	36.28 ± 0.53 ^b^	9.82 ± 0.11 ^d^	6.78 ± 0.19 ^h^	14.54 ± 0.14 ^c^	2.51 ± 0.06 ^d^	4.23 ± 0.06 ^f^	3.01 ± 0.09 ^d e^	6.53 ± 0.07 ^d e^	8.15 ± 0.10 ^a^	11.43 ± 0.08 ^b^	9.90 ± 0.19 ^c d^	8.22 ± 0.11 ^c^
A3	18.56 ± 0.19 ^c^	1.92 ± 0.01 ^f^	1.89 ± 0.02 ^d^	31.95 ± 0.34 ^d^	6.86 ± 0.10 ^g^	6.94 ± 0.17 ^h^	11.07 ± 0.08 ^d^	2.43 ± 0.08 ^d^	3.81 ± 0.17 ^g h^	4.34 ± 0.10 ^b^	7.50 ± 0.20 ^b^	7.34 ± 0.11 ^b^	10.03 ± 0.13 ^d^	10.56 ± 0.15 ^b c^	8.70 ± 0.02 ^b^
A4	8.91 ± 0.59 ^m^	3.32 ± 0.32 ^b^	1.49 ± 0.45 ^g^	34.36 ± 2.31 ^c^	6.45 ± 0.17 ^h^	8.83 ± 0.06 ^e^	16.50 ± 0.45 ^b^	4.13 ± 0.36 ^b^	3.58 ± 0.08 ^i^	3.77 ± 0.10 ^c^	8.05 ± 0.11 ^a^	6.85 ± 0.09 ^c^	10.52 ± 0.21 ^c^	9.97 ± 0.40 ^c d^	8.73 ± 0.17 ^b^
A5	14.54 ± 0.15 ^f^	3.80 ± 0.28 ^a^	1.79 ± 0.02 ^e^	33.62 ± 0.38 ^c^	7.95 ± 0.14 ^f^	5.62 ± 0.28 ^k^	21.06 ± 0.21 ^a^	5.16 ± 0.38 ^a^	3.49 ± 0.18 ^i^	3.62 ± 0.12 ^c^	7.38 ± 0.08 ^b^	6.91 ± 0.09 ^c^	12.02 ± 0.19 ^a^	9.31 ± 0.17 ^e^	8.98 ± 0.14 ^b^
A6	13.58 ± 0.02 ^g^	2.79 ± 0.25 ^d^	1.97 ± 0.04 ^c^	45.13 ± 0.06 ^a^	8.79 ± 0.18 ^e^	6.90 ± 0.02 ^h^	16.68 ± 0.37 ^b^	3.12 ± 0.17 ^c^	4.50 ± 0.05 ^e^	3.57 ± 0.07 ^c^	6.39 ± 0.22 ^e^	6.41 ± 0.17 ^d^	10.32 ± 0.27 ^c d^	9.71 ± 0.14 ^c d^	9.00 ± 0.13 ^b^
B1	10.94 ± 0.11 ^l^	3.11 ± 0.06 ^c^	1.79 ± 0.01 ^e^	19.97 ± 0.23 ^h^	7.88 ± 0.02 ^f^	6.42 ± 0.38 ^j^	5.39 ± 0.28 ^k^	1.26 ± 0.04 ^g^	5.08 ± 0.05 ^c^	2.81 ± 0.14 ^e^	6.55 ± 0.18 ^d^	7.31 ± 0.18 ^b^	8.32 ± 0.20 ^f g^	10.15 ± 0.09 ^c^	8.83 ± 0.24 ^b^
B2	10.80 ± 0.08 ^l^	3.57 ± 0.13 ^a^	2.99 ± 0.04 ^a^	20.81 ± 0.74 ^h^	13.55 ± 0.62 ^a^	7.67 ± 0.14 ^g^	6.63 ± 0.08 ^i^	1.59 ± 0.14 ^e f^	4.35 ± 0.10 ^f^	3.10 ± 0.18 ^d^	6.89 ± 0.27 ^c^	6.95 ± 0.26 ^c^	9.48 ± 0.22 ^e^	8.36 ± 0.36 ^f g^	7.89 ± 0.16 ^d^
B3	8.87 ± 0.62 ^m^	1.37 ± 0.02 ^j^	1.43 ± 0.02 ^g^	23.63 ± 2.23 ^g^	11.64 ± 0.13 ^b^	5.79 ± 0.06 ^k^	7.35 ± 0.21 ^h^	1.77 ± 0.04 ^e f^	3.70 ± 0.07 ^h^	2.54 ± 0.09 ^f^	6.94 ± 0.08 ^c^	6.97 ± 0.17 ^c^	10.13 ± 0.20 ^c d^	10.11 ± 0.28 ^c^	9.92 ± 0.25 ^a^
B4	8.24 ± 0.02 ^m^	3.06 ± 0.08 ^c^	1.66 ± 0.09 ^f^	22.85 ± 0.15 ^g^	10.63 ± 0.57 ^c^	4.73 ± 0.21 ^l^	6.59 ± 0.16 ^i^	1.99 ± 0.19 ^e^	4.32 ± 0.18 ^f^	2.51 ± 0.12 ^f^	6.72 ± 0.12 ^c d^	6.97 ± 0.13 ^c^	9.44 ± 0.10 ^e^	10.10 ± 0.43 ^c^	8.17 ± 0.15 ^c d^
B5	14.21 ± 0.19 ^f^	1.93 ± 0.11 ^f^	1.34 ± 0.04 ^h^	26.06 ± 0.28 ^f^	10.96 ± 0.23 ^c^	6.49 ± 0.64 ^i^	6.08 ± 0.91 ^j^	1.56 ± 0.21 ^f^	4.53 ± 0.09 ^e^	3.72 ± 0.12 ^c^	5.82 ± 0.22 ^f^	5.38 ± 0.21 ^e^	9.66 ± 0.39 ^e^	9.57 ± 0.44 ^d^	7.38 ± 0.20 ^e^
C1	12.45 ± 0.34 ^i^	1.01 ± 0.23 ^m^	1.62 ± 0.06 ^f^	12.57 ± 0.25 ^m^	5.91 ± 0.73 ^h^	10.22 ± 0.12 ^c^	3.00 ± 0.22 ^n^	2.49 ± 0.42 ^d^	4.95 ± 0.16 ^c d^	2.99 ± 0.26 ^d e^	5.01 ± 0.16 ^h i^	4.83 ± 0.07 ^f g^	6.54 ± 0.23 ^i^	7.44 ± 0.26 ^h^	6.04 ± 0.18 ^f^
C2	13.82 ± 0.06 ^g^	1.61 ± 0.02 ^h^	1.30 ± 0.13 ^h^	14.16 ± 0.04 ^k l^	8.55 ± 0.02 ^e^	10.10 ± 0.02 ^c^	7.40 ± 0.12 ^h^	0.41 ± 0.11 ^k^	5.31 ± 0.21 ^b c^	2.03 ± 0.27 ^h^	4.38 ± 0.23 ^j^	5.23 ± 0.12 ^e f^	7.52 ± 0.11 ^h^	7.97 ± 0.16 ^g^	5.64 ± 0.02 ^g^
C3	13.13 ± 0.87 ^h^	0.83 ± 0.11 ^n^	2.64 ± 0.11 ^b^	13.58 ± 0.36 ^l^	7.08 ± 0.08 ^g^	10.47 ± 0.36 ^c^	9.56 ± 0.25 ^f^	0.74 ± 0.02 ^j^	4.71 ± 0.14 ^d^	2.21 ± 0.11 ^g^	4.67 ± 0.22 ^i^	5.04 ± 0.19 ^f^	8.55 ± 0.29 ^f^	7.69 ± 0.17 ^g h^	8.86 ± 0.19 ^b^
C4	14.64 ± 0.17 ^f^	1.26 ± 0.04 ^k^	1.73 ± 0.04 ^e^	15.21 ± 0.21 ^j^	6.32 ± 0.06 ^h^	8.79 ± 0.19 ^e^	3.18 ± 0.10 ^m^	3.87 ± 0.13 ^b^	4.79 ± 0.23 ^d^	1.69 ± 0.13 ^i j^	5.47 ± 0.18 ^g^	4.40 ± 0.11 ^h^	8.06 ± 0.14 ^g^	8.51 ± 0.12 ^f^	5.63 ± 0.15 ^g^
C5	14.22 ± 0.89 ^f^	1.71 ± 0.09 ^g^	1.33 ± 0.04 ^h^	14.75 ± 0.19 ^k^	9.83 ± 0.13 ^d^	13.07 ± 0.11 ^a^	4.07 ± 0.12 ^l^	0.39 ± 0.06 ^k^	4.93 ± 0.12 ^c d^	2.21 ± 0.11 ^g^	3.70 ± 0.21 ^l^	4.28 ± 0.14 ^i^	7.43 ± 0.27 ^h^	6.40 ± 0.29 ^i^	4.89 ± 0.23 ^i l^
C6	11.85 ± 0.30 ^k^	1.83 ± 0.02 ^f^	0.98 ± 0.02 ^i^	13.47 ± 0.34 ^l^	8.41 ± 0.02 ^e^	11.72 ± 0.15 ^b^	5.00 ± 0.03 ^k^	4.11 ± 0.34 ^b^	3.91 ± 0.20 ^g^	2.26 ± 0.11 ^g^	4.01 ± 0.19 ^k^	4.08 ± 0.16 ^i j^	8.53 ± 0.25 ^f^	7.50 ± 0.28 ^g h^	5.36 ± 0.11 ^g h^
C7	12.86 ± 0.19 ^i^	0.76 ± 0.04 ^n^	2.46 ± 0.04 ^b c^	12.27 ± 0.68 ^m^	5.44 ± 0.11 ^i^	9.87 ± 0.02 ^d^	9.83 ± 0.15 ^f^	0.44 ± 0.06 ^k^	5.35 ± 0.18 ^b c^	2.38 ± 0.14 ^f g^	4.06 ± 0.09 ^k^	4.38 ± 0.04 ^h^	7.13 ± 0.18 ^h^	8.44 ± 0.14 ^f g^	5.10 ± 0.15 ^i^
D1	16.73 ± 0.23 ^d^	0.29 ± 0.02 ^o^	1.85 ± 0.04 ^d^	16.83 ± 0.13 ^i^	9.59 ± 0.12 ^d^	8.31 ± 0.02 ^f^	8.19 ± 0.14 ^g^	1.83 ± 0.04 ^e f^	5.87 ± 0.22 ^a^	1.76 ± 0.22 ^i^	5.15 ± 0.21 ^h^	4.74 ± 0.09 ^g^	10.12 ± 0.40 ^c d^	10.68 ± 0.43 ^b^	5.96 ± 0.32 ^f^
D2	18.56 ± 0.23 ^c^	1.56 ± 0.02 ^h i^	1.89 ± 0.15 ^d^	16.65 ± 0.38 ^i^	8.55 ± 0.06 ^e^	9.10 ± 0.17 ^e^	7.58 ± 0.15 ^h^	0.26 ± 0.03 ^l^	4.54 ± 0.25 ^e^	1.30 ± 0.25 ^j^	4.29 ± 0.24 ^j k^	4.56 ± 0.15 ^g h^	8.37 ± 0.31 ^f g^	7.82 ± 0.36 ^g h^	4.63 ± 0.12 ^l^
D3	15.69 ± 0.08 ^e^	0.45 ± 0.02 ^o^	1.81 ± 0.04 ^d^	15.35 ± 0.11 ^j^	7.46 ± 0.18 ^g^	7.22 ± 0.04 ^g h^	8.34 ± 0.15 ^g^	0.29 ± 0.02 ^l^	5.50 ± 0.15 ^b^	2.18 ± 0.14 ^g^	4.76 ± 0.09 ^i^	5.20 ± 0.12 ^e f^	9.63 ± 0.27 ^e^	9.84 ± 0.27 ^c d^	4.58 ± 0.22 ^l^
E1	15.90 ± 0.21 ^d e^	1.48 ± 0.42 ^i^	0.83 ± 0.04 ^j^	10.29 ± 0.34 ^o^	5.34 ± 0.04 ^i^	6.54 ± 0.13 ^i^	10.58 ± 0.37 ^e^	0.83 ± 0.09 ^i^	3.88 ± 0.31 ^g h^	2.13 ± 0.09 ^g^	3.23 ± 0.10 ^m^	3.83 ± 0.26 ^j^	5.47 ± 0.18 ^j^	5.00 ± 0.20 ^j^	3.49 ± 0.14 ^n^
E2	14.25 ± 1.19 ^f^	1.89 ± 0.04 ^f^	1.85 ± 0.12 ^d^	11.13 ± 0.12 ^n^	9.66 ± 0.71 ^d^	8.86 ± 0.49 ^e^	11.37 ± 0.13 ^d^	0.68 ± 0.13 ^j^	5.42 ± 0.10 ^b^	2.03 ± 0.12 ^h^	4.26 ± 0.13 ^j k^	3.51 ± 0.12 ^k^	6.16 ± 0.20 ^i^	4.48 ± 0.17 ^k^	4.16 ± 0.17 ^m^
E3	6.03 ± 0.08 ^n^	1.05 ± 0.08 ^m^	0.97 ± 0.04 ^i^	9.60 ± 0.42 ^o^	6.44 ± 0.16 ^h^	6.99 ± 0.25 ^h^	2.64 ± 0.30 ^n^	1.16 ± 0.11 ^g h^	3.82 ± 0.24 ^g h^	1.55 ± 0.10 ^i j^	3.66 ± 0.11 ^l^	4.02 ± 0.14 ^i j^	5.34 ± 0.15 ^j^	4.55 ± 0.18 ^k^	4.60 ± 0.15 ^l^
E4	14.30 ± 0.11 ^f^	1.16 ± 0.06 ^l^	0.66 ± 0.04 ^k^	7.55 ± 0.42 ^p^	4.65 ± 0.16 ^j^	4.51 ± 0.04 ^l^	2.64 ± 0.28 ^n^	0.98 ± 0.06 ^h^	3.48 ± 0.26 ^i^	1.39 ± 0.20 ^j^	3.31 ± 0.16 ^m^	3.10 ± 0.13 ^l^	6.35 ± 0.16 ^i^	6.46 ± 0.32 ^i^	5.19 ± 0.13 ^h^
E5	12.44 ± 0.06 ^j^	2.16 ± 0.08 ^e^	1.11 ± 0.06 ^i^	8.61 ± 0.02 ^p^	9.59 ± 0.12 ^d^	3.28 ± 0.19 ^m^	3.86 ± 0.06 ^l^	1.01 ± 0.07 ^g h^	2.69 ± 0.13 ^j^	1.33 ± 0.12 ^j^	3.47 ± 0.22 ^m^	3.27 ± 0.09 ^k l^	5.45 ± 0.21 ^j^	6.23 ± 0.15 ^i^	4.64 ± 0.25 ^l^

NOTE: All units are mg/g. Values are mean ± SD (Standard deviation) (*n* = 3). Means with different letters within a column were significantly different (*p* < 0.05). Y_HY_—hyperoside, Y_LU_—luteolin, Y_KP_—kaempferol, Y_QI_—quercitrin, Y_C_—catechin, Y_RU_—rutin, Y_CA_—chlorogenic acid, Y_QU_—quercetin, Y_HE_—hesperetin, Y_AP_—apigenin, Y_PH_—peonidin O-hexoside, Y_PG_—peonidin 3-O-glucoside, Y_CGC_—cyanidin 3-O-glucoside, Y_CSA_—cyanidin O-syringic acid, Y_CGT_—cyanidin 3-O-galactoside.

**Table 2 foods-11-02539-t002:** Pearson correlation coefficients between color value, flavonoids and climate factors.

Flavonoids	L*	a*	b*	X_AMT_	X_AMAT_	X_AMIT_	X_RH_	X_AAP_	X_MW_	X_MAW_	X_EW_	X_ASD_	X_ASP_
Y_HY_	0.253	0.355	−0.111	−0.568 **	−0.555 **	−0.581 **	−0.720 **	−0.613 **	0.488 *	0.461 *	0.468 *	0.495 *	0.506 **
Y_LU_	0.318	0.523 **	−0.536 **	0.022	−0.115	0.141	0.097	0.225	0.051	0.327	0.272	−0.366	−0.319
Y_KP_	0.238	0.379	−0.174	−0.059	−0.045	−0.043	−0.340	−0.195	0.062	0.142	0.083	−0.090	−0.065
Y_QI_	0.582 **	0.727 **	−0.775 **	−0.365	−0.483 *	−0.228	−0.152	−0.153	0.325	0.403 *	0.170	−0.207	−0.109
Y_C_	0.406 *	0.420 *	−0.191	−0.229	−0.166	−0.185	0.040	−0.122	−0.098	−0.013	−0.098	−0.010	0.006
Y_RU_	−0.148	−0.274	0.138	0.129	0.276	0.030	−0.029	−0.299	0.292	0.159	0.042	0.239	0.205
Y_CA_	0.360	0.458 *	−0.469 *	−0.048	−0.176	0.054	−0.071	0.121	0.311	0.400 *	0.259	−0.303	−0.227
Y_QU_	0.323	0.374	−0.504 **	0.086	−0.048	0.192	0.144	0.077	0.238	0.244	0.083	−0.375	−0.345
Y_HE_	−0.205	−0.265	0.216	0.074	0.212	−0.006	−0.062	−0.169	0.024	−0.070	−0.208	0.138	0.130
Y_AP_	0.634 **	0.775 **	−0.563 **	−0.362	−0.470 *	−0.233	−0.244	−0.218	0.301	0.347	0.138	−0.173	−0.131
Y_PH_	0.678 **	0.872 **	−0.612 **	−0.297	−0.428 *	−0.136	−0.115	−0.165	0.145	0.377	0.210	−0.316	−0.275
Y_PG_	0.750 **	0.812 **	−0.585 **	−0.472 *	−0.577 **	−0.332	−0.324	−0.335	0.221	0.433 *	0.287	−0.125	−0.082
Y_CGC_	0.721 **	0.881 **	−0.570 **	−0.417 *	−0.476 *	−0.293	−0.271	−0.390 *	0.228	0.395 *	0.256	−0.039	0.008
Y_CSA_	0.710 **	0.758 **	−0.573 **	−0.502 **	−0.559 **	−0.384	−0.311	−0.510 **	0.162	0.332	0.184	0.034	0.072
Y_CGT_	0.595 **	0.769 **	−0.579 **	−0.301	−0.435 *	−0.147	−0.026	−0.192	0.078	0.281	0.119	−0.235	−0.181

Note: Y_HY_, Y_LU_, Y_KP_, Y_QI_, Y_C_, Y_RU_, Y_CA_, Y_QU_, Y_HE_, Y_AP_, Y_PH_, Y_PG_, Y_CGC_, Y_CSA_ and Y_CGT_ are shown in Table 1. X_AMT_—annual mean temperature (°C), X_AMAT_—annual mean maximum temperature (°C),—annual mean minimum temperature X_AMIT_ (°C), X_RH_—annual relative humidity (%), X_AAP_—annual average precipitation (mm), X_MW_—mean wind speed (m/s), X_MAW_—maximum wind speed (m/s), X_EW_—extreme wind speed (m/s), X_ASD_—annual sunshine duration (h) and X_ASP_—percentage of sunshine (%). ** represents significant correlation at *p* < 0.01 level, * represents significant correlation at *p* < 0.05 level.

**Table 3 foods-11-02539-t003:** Path analysis between climate factors and flavonoids of Chinese prickly ash peels.

Item	Factors	Correlation	Direct Path	Indirect Path Coefficient					Decision	Significance Level
		Coefficients	Coefficients						Coefficient	*p*-Value
Y_HY_										
	X_RH_	−0.720	−0.720						0.518	0.000
Y_LU_				Total	→X_EW_	→X_ASD_	→X_ASP_			
	X_EW_	0.272	0.567	−0.295		−1.148	0.853		−0.013	0.000
	X_ASD_	−0.366	−2.814	2.448	0.230		2.218		−5.859	0.002
	X_ASP_	−0.319	2.245	−2.564	0.216	−2.780			−6.470	0.032
Y_QI_				Total	→X_AMAT_	→X_ASD_	→X_ASP_	→X_MAW_		
	X_AMAT_	−0.483	−0.492	0.009		1.803	−1.673	−0.121	0.233	0.000
	X_ASD_	−0.207	−3.724	3.517	0.237		3.196	0.084	−12.328	0.000
	X_ASP_	−0.109	3.235	−3.344	0.255	−3.678		0.079	−11.172	0.000
	X_MAW_	0.403	0.392	0.011	0.152	−0.794	0.654		0.162	0.001
Y_CA_				Total	→X_MAW_	→X_ASD_	→X_ASP_			
	X_MAW_	0.400	0.516	−0.116		−0.747	0.631		0.147	0.002
	X_ASD_	−0.303	−3.506	3.203	0.110		3.093		−10.166	0.001
	X_ASP_	−0.227	3.132	−3.359	0.104	−3.463			−11.229	0.002
Y_QU_				Total	→X_AMT_	→X_AMIT_				
	X_AMT_	0.086	−2.589	2.675		2.675			−7.149	0.007
	X_AMIT_	0.192	2.730	−2.538	−0.538				−6.403	0.005
Y_AP_				Total	→X_AMAT_	→X_ASD_				
	X_AMAT_	−0.470	−0.723	0.253		0.253			0.157	0.001
	X_ASD_	−0.173	−0.523	0.350	0.350				−0.093	0.008
Y_CGC_										
	X_AMAT_	−0.476	−0.476						0.227	0.014
Y_CSA_										
	X_AMAT_	−0.559	−0.559						0.312	0.003
Y_CGT_				Total	→X_AMIT_	→X_AMT_				
	X_AMIT_	−0.147	3.763	−3.910		−3.910			−15.268	0.000
	X_AMT_	−0.301	−3.989	3.688	3.688				−13.513	0.000
Y_PH_				Total	→X_AMAT_	→X_ASD_	→X_MAW_	→X_MW_		
	X_AMAT_	−0.428	−0.659	0.231		0.339	−0.220	0.112	0.130	0.000
	X_ASD_	−0.316	−0.700	0.384	0.320		0.152	−0.088	−0.047	0.000
	X_MAW_	0.377	0.712	−0.335	0.203	−0.149		−0.389	0.030	0.003
	X_MW_	0.145	−0.470	0.615	0.157	−0.130	0.588		−0.358	0.037
Y_PG_				Total	→X_AMAT_	→X_ASD_	→X_MAW_			
	X_AMAT_	−0.577	−0.746	0.169		0.268	−0.099		0.304	0.000
	X_ASD_	−0.125	−0.554	0.429	0.360		0.069		−0.169	0.001
	X_MAW_	0.433	0.321	0.112	0.230	−0.118			0.175	0.027
a*				Total	→X_AMAT_	→X_ASP_	→X_MAW_			
	X_AMAT_	−0.541	−0.763	0.222		0.323	−0.101		0.243	0.000
	X_ASP_	−0.165	−0.625	0.460	0.394		0.066		−0.185	0.000
	X_MAW_	0.438	0.329	0.109	0.235	−0.126			0.180	0.020
L*										
	X_AMAT_	−0.729	−0.729						0.531	0.001
b*				Total	→X_AMAT_	→X_AMT_	→X_AAP_			
	X_AMAT_	0.505	1.149	−0.644		−0.848	0.204		−0.160	0.006
	X_AMT_	0.567	−0.611	1.178	1.026		0.152		−1.066	0.005
	X_AAP_	0.682	0.502	0.180	0.468	−0.288			0.433	0.010

Note: Y_HY_, Y_LU_, Y_KP_, Y_QI_, Y_C_, Y_RU_, Y_CA_, Y_QU_, Y_HE_, Y_AP_, Y_PH_, Y_PG_, Y_CGC_, Y_CSA_ and Y_CGT_ are listed in Table 1. X_AMT_, X_AMAT_, X_AMIT_, X_RH_, X_MW_, X_MAW_, X_EW_, X_ASD_ and X_ASP_ are listed in Table 2.

**Table 4 foods-11-02539-t004:** The regression equation between color values and flavonoids.

Compounds	Regression	R	R^2^	F
Y_QI_	y = −0.854a* + 0.383b* + 77.186	0.891	0.795	44.511
Y_AP_	y = 0.050a* − 0.042b* + 1.589	0.818	0.668	23.180
Y_PH_	y = 0.088a* − 0.067b* + 2.719	0.849	0.722	57.247
Y_PG_	y = 0.093a* − 0.059b* + 1.376	0.808	0.653	56.694
Y_CGC_	y = 0.111a* − 0.084b* + 5.379	0.913	0.833	29.806
Y_CSA_	y = 0.099a* − 0.092b* + 7.606	0.912	0.831	21.646
Y_CGT_	y = 0.098a* − 0.091b* + 5.709	0.819	0.671	23.420

## Data Availability

The datasets generated during and/or analyzed during the current study are available from the corresponding author on reasonable request.

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
