# Peer review of "Evaluating the Impacts of Climate Factors and Flavonoids Content on Chinese Prickly Ash Peel Color Based on HPLC-MS and Structural Equation Model"

_foods, 2022, doi:10.3390/foods11162539_

Round 1
Reviewer 1 Report
Abstract
Some qualitative and quantitative information in the abstract will be of a lot of help.
A key word might be chemometrics
In general, the document needs consistency in many terms such Maxim or Maxim.
Be careful with the use of a - un many words with a prefix anti (e.g., antiaging). Check them all.
What is the point of using Y in all flavonoids if all have it???
Line 43: quercetin is written twice
Line 50: attach. What do you mean?
Line 74: in plural both groups of compounds
Line 139: Mass Spectrometry with M S
Line 168: What is CIE?
Line 172: Give the name of the samples. It is not easy to see them in the Figure (e.g., B3, D3??). Give examples of the most representative samples in each Figure. Figure 3C not easy to see.
Lines 194-196: …. the significantly higher in each sample?? Give information of the samples. In general, the author could mentioned samples (A….E) involved in some correlations.
Figure 3A: Not all flavonoids were included in the analysis. Why?
It would be good to explain the relationship between the nomenclature used for the samples and Figure 1.
Figure 3B: Are A and B geographically close together?
Figure 3C: Make symbols a bit bigger
The Note under Table 2 is not clear……were performed in Table 3, respectively.???
Lines 296-298: Not clear
Line 303: use were not was
Lines 308-309: Rewrite
In Tables, be consistent with the use of the term flavonoids do not use items or compounds, etc.
Line 337: Should be Table 4 not Table 2.
Figure 4: What is SEM?
Related it to Figure 4: Do the authors have the SEM representation for all flavonoids? It can be added as a supplementary figure.
Line 401: Remove the
Discussion
Lines 433-437: If so, many variables affect the Chinese prickly ash the is like if nothing is relevant or over important.
In discussion many times sugar role on the flavonoid’s synthesis importance is mentioned. Why did the author not measure at least ºBrix?
References
Check the reference style used. There are journal names in capital, some with & and some with and. Many other inhomogeneities.
Author Response
What is the point of using Y in all flavonoids if all have it???
Response: Because of the long name of flavonoids, the use of high frequency in the analysis below, in order to facilitate writing and analysis, we decided to use Y to replace it, such as ‘YCGT’ represented Cyanidin 3-O-galactoside.
Line 43: quercetin is written twice
Response: This is a writing error, the ‘quercetin’ should be ‘quercitrin’, and we have modified ‘quercetin’ to ‘quercitrin’.
Line 50: attach. What do you mean?
Response: The ‘attach great importance to’ mean ‘pay high attention to’, and we have modified ‘attach great importance to’ to ‘pay high attention to’.
Line 74: in plural both groups of compounds
Response: We have modified ‘Flavonoids and anthocyanin’ to ‘Flavonoids and anthocyanins’.
Line 139: Mass Spectrometry with M S
Response: MS was the abbreviation of Mass Spectrometry.
Line 168: What is CIE?
Response: CIE was the abbreviation of ‘International Lighting Commission’.
Line 172: Give the name of the samples. It is not easy to see them in the Figure (e.g., B3, D3??). Give examples of the most representative samples in each Figure. Figure 3C not easy to see.
Response: All varieties belong to the Dahongpao in Zanthoxylum bungeanum, but with no specific name. The detailed sampling information on Zanthoxylum bungeanum peels was listed in Table S1.
Lines 194-196: …. the significantly higher in each sample?? Give information of the samples. In general, the author could mentioned samples (A….E) involved in some correlations.
Response: The contents of hyperoside and quercitrin were significantly higher than that of other compounds. All varieties belong to the Dahongpao in Zanthoxylum bungeanum, but with no specific name. The detailed sampling information on Zanthoxylum bungeanum peels was listed in Table S1.
Figure 3A: Not all flavonoids were included in the analysis. Why?
Response: The plot of correlation coefficient analysis between flavonoids content and geographical factors has modified.
It would be good to explain the relationship between the nomenclature used for the samples and Figure 1.
Response: We have added the information of sampling sites to the map of collection sample sites of 26 Chinese prickly ash peels.
Figure 3B: Are A and B geographically close together?
Response: The geographical location of A and B samples was not close, and they were distributed in western Shaanxi (Fengxian), eastern Qinghai (Xunhua, Guide), Gansu (Wudu, Qinan) and Hanyuan in Sichuan. The samples within A and B were clustered together, indicating that the composition content of these samples was similar.
Figure 3C: Make symbols a bit bigger
Response: We have adjusted the size of symbols in Figure 3C for observation.
The Note under Table 2 is not clear……were performed in Table 3, respectively.???
Response: The Note under Table 2: YHY, YLU, YKP, YQI, YC, YRU, YCA, YQU, YHE, YAP, YPH, YPG, YCGC, YCSA and YCGT were performed in Table 1, respectively. XAMT (°C)-Annual mean temperature, XAMAT (°C)-Annual mean maximum temperature, XAMIT (°C)-Annual mean minimum temperature, XRH (%)-Annual relative humidity, XAAP (mm)-Annual average precipitation, XMW (m/s)-Mean wind speed, XMAW (m/s)-Maximum wind speed, XEW (m/s)-Extreme wind speed, XASD (h)-Annual sunshine duration and XASP (%)-Percentage of sunshine., respectively.
Lines 296-298: Not clear
Response: At the same time, climate factors were used as the dependent variables to carry on analysis. The process was conducted as follows: At first, stepwise regression analysis was carried out on the climate factors, flavonoids and peel color by SPSS statistical software.
Line 303: use were not was
Response: We have modified ‘was’ to ‘were’. The results of PA (Table 3) in this study demonstrated the effects of climate factors were significant on flavonoids compounds contents.
Lines 308-309: Rewrite
Response: XMAW was the key environmental factor for YQI, YPH and YCA, and was present with positive direct effects.
In Tables, be consistent with the use of the term flavonoids do not use items or compounds, etc.
Response: We have modified the use of the term flavonoids in Tables, and made the use of the term flavonoids was consistent.
Line 337: Should be Table 4 not Table 2.
Response: Table 2 was to reveal the correlation between flavonoids content and color values. Table 4 was to reveal regression analysis to explore the quantitative relationship between the content and color value.
Figure 4: What is SEM?
Response: SEM was the abbreviation of Structural equation model. Structural equation model (SEM) offers a means to evaluate hypothesised causal relationships amongst multiple variables.
Related it to Figure 4: Do the authors have the SEM representation for all flavonoids? It can be added as a supplementary figure.
Response: Using sample data for model fitting analysis, CMIN/DF < 1, RMSEA < 0.08 and AGFI > 0.9 indicated that the model was excellent with reasonable adaptation. Therefore, the models with CMIN/DF < 1, RMSEA < 0.08 and AGFI > 0.9 were selected. Finally, what we selected was described in Figure 4.
Line 401: Remove the
Response: We have removed the ‘the’.
Discussion
Lines 433-437: If so, many variables affect the Chinese prickly ash the is like if nothing is relevant or over important.
Response: Climate factors have interaction, directly or indirectly affect the content of flavonoids in Chinese prickly ash peels peel, and each climatic factor plays a different role in flavonoids accumulation.
Reviewer 2 Report
In the manuscript "Evaluating the impacts of climate factors and flavonoids compounds on Chinese prickly ash peels color based on HPLC-MS and structural equation model" the authors presented an interesting approach for analysis of climatic factors on flavonoids accumulation. Although interesting, the manuscript has some important backdraws that should be explained. My biggest concern goes to the discussion of the climate factor based on just one year of meteorological data. Those data should be expanded on for a few years before discussing the climate effect. The authors should explain that in detail. My suggestion is a major revision, according to the listed comments:
Reference number 1 was not mentioned in the manuscript.
Line 128. Add information about the producer of the colorimeter
Line 147. Add information on how the results of the HPLC-MS analysis were presented.
Line 155. Data in table S2 should be presented with data dispersion.
Please checks the tables numbering in supplementary materials.
Line 183-184. "Moreover, differences within groups were small, while differences among groups were significant." Which test was used for the statistical analysis and how were the differences presented in Figure 2?
Figure 2. Why did the authors use a radar chart for results presentation? In my opinion, the bar chart would be more suitable.
There is no comparison of the colour results with literature data.
Line 220. Why were Pearson correlation coefficients used? Did the authors test the normality of the data prior to the correlation analysis?
Figure 3B. Please include the explanation of the groups in the legend or in the figure captions.
Line 233. The authors mentioned samples grouping into four grouping according to PCA, but there are five groups presented at Figure 3B! Please explain the data grouping.
Line 241. How was normalisation performed?
Line 279. Average values or all parallels were used for correlation analysis?
Table 4. Is should be "regression equation" instead of "regressive"
Author Response
Reference number 1 was not mentioned in the manuscript.
Response: We have replaced Reference Number 1 with another reference.
Line 128. Add information about the producer of the colorimeter
Response: We have added information about the producer of the colorimeter, such as NH310 computer colorimeter (Shenzhen ThreeNH Technology Co., Ltd.).
Line 147. Add information on how the results of the HPLC-MS analysis were presented.
Response: Flavonoids data analysis were analyzed by Analyst 1.6.3 software (AB SCIEX, Concord, Ont, Canada). Integration and correction of chromatographic peaks were performed using MultiQuant 3.02 (AB SCIEX, Concord, Ontario, Canada). The corre-sponding relative flavonoids content was expressed as the chromatographic peak area integral.
Line 155. Data in table S2 should be presented with data dispersion.
Response: We have checked and modified the Table S2, in order of Table S1.
Please checks the tables numbering in supplementary materials.
Response: We have checked and modified the tables numbering in supplementary materials.
Line 183-184. "Moreover, differences within groups were small, while differences among groups were significant." Which test was used for the statistical analysis and how were the differences presented in Figure 2?
Response: Therefore, the Chinese prickly ash peels color from different habitats had regional differences.
Figure 2. Why did the authors use a radar chart for results presentation? In my opinion, the bar chart would be more suitable.
Response: We have redrawn the color data of Chinese prickly ash peels, and the radar chart was converted to bar chart.
There is no comparison of the colour results with literature data.
Response: The color of Chinese prickly ash peels was controlled by climate, flavonoids and anthocyanidins, and the difference in climate caused the difference in peel color. Climate factors in each year was different, and Chinese prickly ash peels color was not the same.
Line 220. Why were Pearson correlation coefficients used? Did the authors test the normality of the data prior to the correlation analysis?
Response: Pearson correlation coefficients were to explore the correlations between flavonoids and geographic factor of sampling sites. We have tested the normality of the data before conducting the correlation analysis.
Figure 3B. Please include the explanation of the groups in the legend or in the figure captions.
Response: PCA on 2-D plot of Chinese prickly ash based on flavonoids compounds contents. Four different color ellipses represented PCA results.
Line 233. The authors mentioned samples grouping into four grouping according to PCA, but there are five groups presented at Figure 3B! Please explain the data grouping.
Response: Samples were divided into four grouping according to PCA, and four different color ellipses represent four groups.
Line 241. How was normalisation performed?
Response: All data are normalized by Unit Variance Scaling.
Line 279. Average values or all parallels were used for correlation analysis?
Response: Average values were used for correlation analysis?
Table 4. Is should be "regression equation" instead of "regressive"
Response: We have modified ‘regressive equation’ to ‘regression equation’, such as ‘The regression equation between color values and flavonoids’.
Reviewer 3 Report
This paper aimed to apply the strategy based on HPLC-MS technology and structural equation model to investigate the effects of climate factors and flavonoids content on Chinese prickly ash peels color. Different environmental factors were investigated in relation to the flavonoids content and peel color. The correlation models of climatic factors and flavonoids with peel color were established based on HPLC-MS technology and structural equation model. Special attention was devoted to collection of samples.
I suggest for publication after minor revision. Review is in attach.

Author Response
Lines 19-24: This sentence should be divided into two or more sentences, to achieve easier reading.
Response: This is the significance of our research, all of which are obtained from the experimental results. We think it is more appropriate to write them together.
The title of the paper adequately reflects the subject under investigation in the proposed study, but might be slightly modifed to: „Evaluating the impacts of climate factors and flavonoids content on Chinese prickly ash peels color based on HPLC-MS and structural equation model“
Response: We have carefully checked and modified the title to ‘Evaluating the impacts of climate factors and flavonoids content on Chinese prickly ash peels color based on HPLC-MS and structural equation model’.
Line 25: flavonoids content
Response: We have carefully checked the keywords and modified the ‘flavonoids compounds’ to ‘flavonoids content’
References are numbered in order of appearance in the text, as demanded by formatting rules of the journal, but it shoud be checked the corectness of citations in the text.
Response: We have replaced Reference Number 1 with another reference.
Line 33: change one of the words „different“ with some synonym
Response: The Chinese prickly ash peel color produced in different regions has difference.
Line 64: Please define terms of ground color and cover color
Response: The color (yellow) shown by carotenoids and flavonoids forms the ground color of the fruit surface and the color (red) shown by anthocyanin forms the cover color.
Line 78: Please check the correctness od reference citation
Response: Katerina Biniari found that the accumulation of flavonoids in grape skins and seeds were controlled and regulated by air temperature and wind speed
Line 132: Please identify how many replications of samples were made
Response: All samples were made in three replicates.
Line 135: Add the name of apparatus for deionized water production and the producer name and location.
Response: Deionized water (18 MΩ cm) was used to prepare aqueous solutions (MilH-Q Ad-vantage A1, Millipore, USA).
Line 136: Please specify which membrane filters
Response: The supernatant was filtered by microporous membrane (SCAA-104, 0.22 µm pore size, ANPEL, Shanghai, China) and the filtrates were stored in the injection bottle.
Lines 139-140: Please add some more details about HPLC-MS apparatus (pump, degasser...), producer name etc.
Response: The quantitative analysis of 15 flavonoids compounds in Chinese prickly ash peels was carried out based on HPLC-MS (HPLC, Shim-pack UFLC SHIMADZU CBM30A system; MS, Applied Biosystems 4500 Q TRAP) technology.
Lines 142-143: Please identify which percentage corresponds to which mobile phase in the gradient mode.
Response: Solvent system: solvent A, water (0.04% acetic acid), solvent B, acetonitrile (0.04% acetic acid); gradient program, 95:5 v/v (A/B) at 0min, 5:95 v/v at 11.0min, A:95 v/v at 12.0min, 95:5 v/v at 12.1min, 95:5 v/v at 15.0 min.
Line 146: Please identify the ESI abbreviation and state what software was used to control it.
Response: ESI was the abbreviation of electrospray ionization. The effluent was alternatively connected to an ESI-triple quadrupole-linear ion trap (QTRAP)-MS. Metabolite quantification was performed in multiple reaction monitoring (MRM) mode using triple quadrupole mass spectrometry
Line 206-210: Are there any publications with similar findings? If are, please cite some
Response: The color of Chinese prickly ash peels was controlled by climate, flavonoids and anthocyanidins, and the difference in climate caused the difference in peel color. Climate factors in each year was different, and Chinese prickly ash peels color was not the same.
Line 231: Please remove the space between number (48.36) and %, and check this through the whole manuscript.
Response: We have carefully checked and removed the space between number and % through the whole manuscript.
Line 401: „... in order to quickly evaluate the its quality.“ Please delete „the“
Response: We have removed the ‘the’.
Lines 423: „Olha Mykhailenko and his team found...“ Please check the corectness of citation
Response: Olha Mykhailenko found that flavonoids compounds accumulation of in Iris species was positively regulated by sunshine duration.
Line 431: change was to were
Response: For example, the flavonoids contents in three samples (A3, C4, E2) were varied from each other.
Lines 454-455: „When the light intensity of the fruit was more than 70 % of the natural light intensity, the peel color was excellent.“ Please improve the style.
Response: When the light intensity was more than 70 % of the natural light intensity, the peel color was excellent.
The Language should be edited by a native editor.
Response: The significant issues with the English (syntax, grammar, etc.) throughout the manuscript have been addressed.

Round 2
Reviewer 1 Report
The reviewer is satisfied with the made changes.
Author Response
Thank you for your valuable commentsReviewer 2 Report
The authors put an effort and answered the comments and suggestions. In my opinion they have improved the manuscript and therefore I think that the manuscript can be accepted for publication.
Author Response
Thank you for your valuable comments